# Identifying Patterns of Symptom Distress in Pregnant Women: A Pilot Study

**DOI:** 10.3390/ijerph18126333

**Published:** 2021-06-11

**Authors:** Ching-Fang Lee, Fur-Hsing Wen, Yvonne Hsiung, Jian-Pei Huang, Chun-Wei Chang, Hung-Hui Chen

**Affiliations:** 1Department of Nursing, Mackay Medical College, New Taipei City 25245, Taiwan; yvonnebear@mmc.edu.tw; 2Department of International Business, Soochow University, Taipei City 100006, Taiwan; wenft@scu.edu.tw; 3Department of Obstetrics and Gynecology, Mackay Memorial Hospital, Taipei City 104217, Taiwan; huangjianpei@yahoo.com.tw; 4Department of Psychiatry, Fu Jen Catholic University Hospital, New Taipei City 24352, Taiwan; weinavy@ms22.hinet.net; 5School of Nursing, College of Medicine, National Taiwan University, Taipei City 10051, Taiwan; hunghuichen@ntu.edu.tw; 6Department of Nursing, National Taiwan University Hospital, Taipei City 10002, Taiwan

**Keywords:** pregnant women, pregnancy-induced symptom distress, patterns, trajectory

## Abstract

During pregnancy, a woman’s enlarged uterus and the developing fetus lead to symptom distress; in turn, physical and psychological aspects of symptom distress are often associated with adverse prenatal and birth outcomes. This study aimed to identify the trends in the trajectory of these symptoms. This longitudinal study recruited 95 pregnant women, with a mean age of 32 years, from the prenatal wards of two teaching hospitals in northern Taiwan. Symptom distress was measured by a 22-item scale related to pregnancy-induced symptoms. The follow-up measurements began during the first trimester and were taken every two to four weeks until childbirth. More than half of the pregnant women experienced symptom distress manifested in a pattern depicted to be “Decreased then Increased” (56.8%). Other noticeable patterns were “Continuously Increased” (28.4%), “Increased then Decreased” (10.5%) and “Continuously Decreased” (4.2%), respectively. It is worth noting that most pregnant women recorded a transit and increase in their symptom distress, revealed by their total scores, at the second trimester (mean 22.02 weeks) of pregnancy. The participants’ major pregnancy-related distress symptoms were physical and included fatigue, frequent urination, lower back pain, and difficulty sleeping. The mean scores for individual symptoms ranged from 2.32 to 3.61 and were below the “moderately distressful” level. This study provides evidence that could be used to predict women’s pregnancy-related symptom distress and help healthcare providers implement timely interventions to improve prenatal care.

## 1. Introduction

Pregnancy-induced symptoms fluctuate across gestational weeks [1,2]. These symp-toms cause significant distress that impacts daily functioning, interferes with work per-formance and impairs the quality of life of both pregnant women and their families [3,4,5]. Moreover, persistent and severe symptoms are associated with adverse effects regarding fetal development and increased pregnancy risks, such as preeclampsia, vaginal bleeding, preterm birth and postpartum complications [6,7,8].

Most previous studies of pregnant women have focused on a single symptom or ex-amined the association among several symptoms, and how these symptoms are related to birth outcomes or postpartum complications [9,10]. In fact, previous studies have primarily addressed a wide variety of symptoms that women tend to experience only in their third trimester [1], or followed symptoms only from late in the third trimester to the early postpartum period [9,11]. However, the pattern of fluctuation in the severity of symptoms [1,12], including distress levels, differs among pregnant women and has not yet been ex-plored in depth. In particular, pregnant women’s level of perceived distress and their varying degrees of awareness of such distress require greater exploration to create a more precise picture.

Symptom distress is an individual’s perceived frequency of particular symptoms, as well as the severity or the level of distress caused by the symptoms [13]. It is worth noting that for pregnant women, symptom frequency may not be the major cause of distress. To our knowledge, pregnancy-induced symptom distress has not been well studied, and only when a full spectrum of symptoms is explored can prenatal interventions be designed to offer guidance in relieving the distress caused by persistent and severe symptoms during pregnancy. Previous studies have found that professional, individualized health education and support can promptly relieve the severity of some symptoms during pregnancy [14,15]. Therefore, we first need to ascertain the dominant trends in the symptom distress of pregnant women in order to educate them about the most common trimester-specific symptoms they can expect to experience. Moreover, understanding the pattern of pregnancy-induced symptom distress could help healthcare providers guide these women more effectively throughout their pregnancy.

Thus, the aims of this study were (i) to identify the patterns of distress-related preg-nancy-induced symptoms and (ii) to examine symptom distress scores in the context of multipattern trends. Our findings may facilitate the identification of women who display a clear symptom-distress trajectory and may thus enable healthcare providers to more ef-fectively take the appropriate actions.

## 2. Materials and Methods

### 2.1. Design

This pilot study used a longitudinal study design based on convenience sampling.

### 2.2. Participants and Setting

More than 99.9% of women, regardless of their gestation statuses or pregnancy related risks, give birth by obstetricians at hospitals in Taiwan [16]. Moreover, pregnant women who have national health insurance are entitled to 10 government-funded prenatal check-ups by obstetricians or midwives. Pregnant Taiwanese women receive a maternal health booklet once the pregnancy has been confirmed based on the fetal heartbeat. The number of regular check-ups in the first, second and third trimesters is two, two and six, respectively [16].

The participants were recruited from the prenatal outpatient departments with 28 obstetricians from two teaching hospitals in Taipei, Taiwan. Inclusion criteria were women in 8–12 weeks of gestation with a singleton pregnancy who had no obstetric or medical complications. Excluded were those who had high obstetric risks or medical complications, such as insulin-dependent diabetes, gestational diabetes, hypertension and multiple-gestation pregnancies. We used the G-Power software package (version 3.1; Heinrich Heine University Düsseldorf, Dusseldorf, Germany) to calculate the necessary sample size, assuming a power of 80%, the medium effect size of 0.5 by Cohen’s analysis and α = 0.05 [17]. The sample size estimate was calculated at 92 (N) in our study to reach satisfactory statistical power. This number also provided power confidence since the rule of thumb in the longitudinal study design where the sample size at effect size 0.5 was 64 (for power = 0.8 for a 2-tailed 0.05 test) [18].

A total of 98 pregnant women completed the questionnaire. The final sample included 95 women who had completed at least two measurements. The mean number of time points at which the women completed the measurements was 5.29 (SD = 1.91; minimum = 2; maximum = 10). Only 5.3% (*n* = 5) participants completed two measurements.

### 2.3. Data Collection

The study was undertaken from July 2016 until June 2017. Symptom-related measurement began during 8–12 weeks of the first trimester after the pregnancy was confirmed by obstetricians, and at least two subsequent measurements were collected by researchers every 4 weeks in the second trimester and every two weeks in the third trimester during prenatal visits until childbirth.

After the women completed the questionnaires, we gave them each a gift and thanked them for their participation. The participants were scheduled to receive a routine prenatal health check-up from multiple obstetricians in Taiwan. The follow-up questionnaires, handed out to participants face to face, were to be returned by them within 2–4 weeks during one of their regular prenatal check-ups at the hospital. We kept a record of when the questionnaires were given out and returned. We also contacted each participant by phone at least one week before her next appointment.

### 2.4. Measurements

Our study’s variables included the sociodemographic details of the participants (i.e., age, level of education, occupation, body weight, prepregnancy body mass index (BMI) and monthly family income), their perinatal characteristics (i.e., parity, abortion history and gestational age), and any pregnancy-related symptom distress.

The participants were asked to record their pregnancy-related symptom distress over the course of the previous 7 days using a 22-item questionnaire (Mandarin version) with content validity index (CVI) values of 98.6–99.2% and a Cronbach’s α value of 0.86 in pregnant Taiwanese women [19,20]. Of the 22 items, 17 were related to physical symptoms (i.e., fatigue, nausea, vomiting, dizziness, frequent urination, lower back pain, leg cramps, leg swelling, difficulty sleeping, constipation, hemorrhoids, abdominal distention, diarrhea, uterine contractions, shortness of breath, numbness/tingling in hands/feet and lumbopelvic pain), while 5 items pertained to psychological symptoms (i.e., worry, anxiety, depression, feeling upset and feeling unhappy). Each symptom was self-reported, with the perceived level of distress being ranked using a 10-point scale (10 = extreme suffering; 0 = no suffering at all). In aspects of physical distress’ and psychological distress’ mean total distress, the highest possible total score was 220.

Higher scores indicated more severe levels of pregnancy-related distress. The Cronbach’s α values of our study for total, physical and psychological symptom distress were 0.93, 0.90 and 0.96, respectively. The CVI values were 96.25, 97.5% and 95%, respectively.

### 2.5. Data Analyses

Aim 1: To identify the patterns of distress-related pregnancy-induced symptoms.

We estimated the trend in the distress scores by hierarchical linear modelling (HLM). Because each participant had asymmetric time points of symptom distress scores in the data-collecting period, the HLM permits the repeated measures within a nested structure [21]. The nested structure included the repeated measures of the distress scores at the within-subject level and at the between-subjects level.

Comparison of the linear and quadratic growth curve models showed that the final model was significantly related to the quadratic growth curve model (χ^2^ = 22.06, df = 4, *p* = 0.001). Based on quadratic growth model, we estimated the trend and pattern of change. Three classes of parameters, i.e., intercept coefficients (the values of the dependent variables in the early weeks of gestation), linear slope coefficients (to estimate the slopes of the tangent lines for the dependent variables in the early weeks of gestation) and curve slope coefficients (to estimate the curvature of the dependent variables’ trajectories during pregnancy), were estimated via the empirical Bayes method to obtain the quadratic curve of the growth model for all participants [22,23]. The coefficient values of the linear slope and curve slope and the times of the estimated maximal and minimal symptom distress scores were used to characterize the patterns of change in the scores. The following four patterns were identified: “Decreased then Increased”, “Increased then Decreased”, “Continuously Increased” and “Continuously Decreased”.

Aim 2: To examine symptom distress scores in the context of multipattern trends.

Statistics were organized into specific aims by means of SPSS 20.0 (SPSS, Chicago, IL, USA). The demographic and prenatal variables were described in terms of frequencies and percentages, while the continuous variables were described in terms of means and standard deviations. To compare score differences among the four identified patterns, one-way ANOVA was used to explore the distributions and compare the means of total, physical, psychological and total average symptom distress, applying a post hoc Scheffe test. The mean values of the total average score were used to determine the distress levels in the four patterns based on a 10-point scale. The total average scores were the summed scores of individual distress divided by the number of symptoms. The distribution and severity of individual symptom distress were compared by ANOVA to the means among the four patterns.

### 2.6. Ethical Considerations

This study was approved by the Institutional Review Board of Mackay Medical Hospital in Taipei, Taiwan (#16MMHIS135). The consent forms signed by the participants included both a brief description of the study and an explicit assurance of confidentiality.

## 3. Results

### 3.1. Participant Characteristics

The final sample was 95 pregnant women whose average age was 32 years (SD = 3.91). The majority had a college or university education (*n* = 87, 91.6%) and a monthly family income of NT 60,000–100,000 (*n* = 54, 56.9%), were employed (*n* = 82, 86.3%), were nulliparous (*n* = 52, 54.7%), had no abortion history (*n* = 84, 88.4%) and had had a normal prepregnancy BMI (*n* = 82, 86.3%). The mean gestational weight gain was 9.11 kg (SD = 4.22) (see Table 1).

### 3.2. Patterns and Trends in Pregnancy-Induced Symptom Distress

The most common pattern was “Decreased then Increased” (*n* = 54, 56.8%); the second most common pattern was “Continuously Increased” (*n* = 27, 28.4%), followed by “Increased then Decreased” (*n* = 10, 10.5%) and “Continuously Decreased” (*n* = 4, 4.2%) (Table 2). Moreover, the total distress scores for pregnancy-induced symptom distress in the “Decreased then Increased” pattern (mean = 51.03) were significantly lower than those in the “Continuously Increased” pattern (mean = 66.20) and the “Continuously Decreased” (mean = 79.41) pattern (Table 2).

The point of transition, when symptom distress scores began to increase, came at 22.02 weeks of gestation for the “Decreased then Increased” pattern. For the “Increased then Decreased” pattern, pregnancy-related symptoms began to decrease at 26.77 weeks (see Figure 1).

### 3.3. The Distribution of Pregnancy-Related Symptom Distress Scores

Table 2 shows that the distress scores for total, physical and psychological symptoms were significantly different for the four patterns (*p* < 0.05). According to the post hoc test, the “Decreased then Increased” pattern had a significantly lower total distress score (mean = 51.03, SE = 3.52), total average distress score (mean = 2.32, SE = 0.16), physical distress score (mean = 41.42, SE = 2.65) and psychological distress score (mean = 9.56, SE = 1.06) than the other patterns, but the “Continuously Decreased” pattern had a higher total distress score (mean = 79.41, SE = 12.83), total average distress score (mean = 3.61, SE = 0.58), physical distress score (mean = 61.10, SE = 9.64) and psychological distress score (mean = 18.22, SE = 2.85). The mean distress scores for total average were below the “moderately distressful” level (mean scores < 5), and the range across all four patterns was 2.32–3.61 (Table 2). For the “Decreased then Increased” trajectory, while the overall pattern changed at 22.02 weeks, physical distress levels increased earlier (21.01 weeks) relative to psychological distress levels (22.34 weeks).

### 3.4. Levels of Distress for Individual Symptoms Across the Patterns

Among pregnancy-related symptoms, fatigue was the worst example of symptom distress in all groups (mean range = 4.06–6.23), with a significantly lower distress level in the “Decreased then Increased” pattern than in the other patterns (*p* < 0.05) (Table 3). Worry was the worst psychological symptom in all the patterns (mean range = 2.36–4.65), but the level of significance was marginal across all four patterns (*p* = 0.05). The distress level scores for individual symptoms—fatigue, difficulty sleeping, uterine contractions, numbness/tingling in hands/feet, diarrhea, anxiety, feeling upset and feeling unhappy—were significantly different for the four patterns (*p* < 0.05) and were significantly lower in the “Decreased then Increased” pattern, but higher scores in the “Continuously Increased” pattern the level of significance were not significant across all four patterns (*p* > 0.05).

## 4. Discussion

Our study found that the “Decreased then Increased” trend displayed the most noticeable changes and had significantly lower symptom distress scores than the other patterns; moreover, in this trend, the point of transition to increasing symptom distress scores came at 22 weeks. We also found that physical symptom distress increased first (21.01 weeks) compared to psychological distress (22.34 weeks) under the most noticeable changing pattern of “Decreased then Increased”. The key cause of increasing distress and physical distress before psychological distress may be related to maternal physiological changes resulting from fetal growth and increasing uterine size at 21–24 gestational weeks [24,25]. Many physiological changes during pregnancy, including the rapid increase in maternal plasma volume to transport nutrients to meet fetal growth demands, rapid weight gain from water retention and fetal growth and significant gravida changes that impact the musculoskeletal system, induce symptom distress after 20 gestational weeks [1,25,26]. Moreover, these physical factors such as anatomy and physical changing had increasing distress level with gestation weeks. The physical symptoms may affect psychological symptoms and drive the psychological distress changes after.

More than 80% of the women in this study perceived increasing symptom distress in the last trimester, especially those exhibiting the “Continuously Increased” (28.4%) and “Decreased then Increased” (56.8%) patterns. This finding suggests that most women continue to experience upward trends and increasingly severe symptoms in their second and third trimesters. When prenatal symptom distress becomes more persistent and severe, women with perinatal distress tend to request extended sick leave for more frequent prenatal care; perceived dissatisfaction with the division of household tasks has also been found [3]. In fact, 85% of women wanted to discuss pregnancy-related symptoms [2], implying a need for healthcare providers to pay special attention to assessing women’s symptom distress, especially if those symptoms are disrupting normal functioning. Providing early counseling to pregnant women was found to be effective in reducing symptom distress [27,28]. For example, during prenatal visits, healthcare providers may detect and assess common pregnancy-related symptoms in the waiting room with a clinically useful checklist and examine patterns or trends in symptom distress by asking simple questions regarding symptom history, occurrence frequency, severity level and life events that may aggravate symptom distress.

Nearly 15% of the women exhibited decreasing patterns of symptoms during pregnancy: “Continuously Decreased” (4.2%) and “Increased then Decreased” (10.5%). Our study also found that women exhibiting the “Increased then Decreased” pattern suffered higher levels of distress than women with the “Continuously Increased” pattern before the turning point toward decreasing symptoms at 26–27 weeks. Furthermore, it is important to note that women who exhibited the “Continuously Decreased” pattern had the highest mean distress score (mean = 3.61). We also found that a change in the pattern of symptom distress was physical before the psychological distress. In order to prevent or relieve physical symptoms, which may affect psychological distress symptom, we could provide education on self-care for the majority of physical symptoms in the first trimester. Early reduction in physical symptom distress may decrease overall symptom distress in these women, which may reflect increased levels of perceived psychological or overall symptom distress. Previous studies have found that symptom distress in the first trimester affects postpartum psychological vulnerability [10,29]. Furthermore, self-care related education about the ways to relieve early physical symptom distress should soon be provided to pregnant women at the first prenatal care visit. More generally, we suggest providing pregnant women detailed educational information about the patterns of changes in symptoms, key transition times and the levels of perceived symptoms they can expect. Women receiving care from healthcare providers reported an increase in knowledgeability about birth and were prepared to use planning-preparation coping strategies to manage the strains and challenges of pregnancy and reduce their fear of childbirth [30]. Therefore, women with early symptom distress should be given early assistance through prenatal education.

Fatigue was the worst symptom of distress in pregnant women in all patterns. This persistent or severe tiredness, resulting in decreased ability to perform daily tasks [3], is related to maternal water retention changes due to increased blood volume and cardio loading during pregnancy [25]. Pregnancy-related physical symptoms such as frequent urination, lower back pain and difficulty sleeping were also commonly mentioned by our subjects, and these were caused by anatomic and physiological changes mediated by fetal growth and uterine enlargement, changes in the center of gravity and pressure on or displacement of the organs [25,26].

Among the psychological distress symptoms, worry and upset had the highest distress scores across the patterns. The physical and psychological symptoms affected each other: the severity of pregnancy-related physical symptoms was strongly associated with worry, and the psychological distress symptom of depression was correlated with fatigue [2]. Women who suffer from persistent back pain and difficulty sleeping may experience fatigue and have psychological problems from pregnancy in the postpartum period [9,12,31]. A recent cohort study of 1858 women by Aukia et al. found that physical complaints may induce sleep difficulty, and that sleep problems and psychological distress symptoms are strongly related [31], indicating the importance of identifying psychological distress among women with persistent or severe physical symptoms throughout pregnancy.

Women tend to feel upset about the uncertain future of the unborn baby and the pregnancy [32], and the co-occurrence of upset emotional states with pregnancy complications raises major concerns [2]. For these women, pregnancy is a risk rather than a cause, and medical attention and symptom relief are necessary. Previous studies have found that professional and individualized health education, counseling and support can promptly relieve the severity or even prevent the appearance of some symptoms during pregnancy [14,15]. Furthermore, education on self-care to help pregnant women relieve pregnancy-induced symptoms could decrease or prevent high levels of psychological distress. Women who perceived more pregnancy distress were at high risk of poor psychological adjustment and self-care [33,34]. Unsuccessful coping with symptom distress not only induces stress [28] but also increases the risks of preterm delivery and low birth weight [6].

We suggest antenatal screening in order to identify potential pregnancy-related symptoms and forms of distress. For example, sleep problems may have serious effects and contribute to adverse pregnancy outcomes or postnatal physical and psychological symptom distress [35,36]. Some symptoms, such as nausea, vomiting, constipation and hemorrhoids, are rarely life-threatening or have adverse consequences but can cause significant distress, not only negatively impacting pregnant women’s quality of life and daily functioning but also leading some women to consider termination of pregnancy or future pregnancy avoidance [37,38]. Once the pattern of symptoms is identified, treatment decisions should be made by specialists.

More generally, the ability of women to identify their pattern of change early and adopt self-care may allow them to prevent or relieve pregnancy-induced symptoms. For example, if a pregnant woman complains about low back pain, self-care strategies may be introduced, such as light yoga exercise, acupressure, lumbar stabilization or stretching exercises, to improve postural stability [39]. Additionally, for multiparous women, family can be encouraged to provide support in childcare or plan for enrollment in a childcare center to decrease fatigue due to caregiving tasks [3,40].

The pregnancy-specific distress experienced by women in the present study was below the moderate level on a 10-point Likert scale (range = 2.32 to 3.61), lower than in the Winkel et al. study conducted in 2013. When we converted their ratings based on a 4-point Likert scale to ratings based on a 10-point scale, the range of the scores for 11 symptoms was 3.0–6.0. However, the subjects of the Winkel et al. study were younger than our subjects (mean age = 28.0 years) and were unemployed and multiparous (50.3%) [41]. Women with lower socioeconomic status and those who are pregnant for the first time tend to experience greater distress than other women [40]. Symptom distress occurs in multiple domains and is affected by personal, behavioral, environmental, social contextual, economic (access to health care) and geographic [13] factors. Beebe et al.’s (2017) study also found that pregnant women’s ethnicity may be correlated with their experience of symptom distress. Asian women tend to have significantly lower symptom distress than the general populations of Euro-Americans, while Latina women tend to have significantly higher symptom distress [1]. This finding may explain why the symptom distress of the pregnant women in our study did not seem very severe overall.

Regarding the clinical implications, these findings can enhance healthcare providers’ detection and appraisal of different patients’ symptoms in the context of clients’ personal and psychosocial environment [42]. Women’s pregnancy-induced symptom distress may be related to their various experiences of partner relationships and family support, coping strategies and socioeconomic status [1,13,30]. Future research should attempt to explore patients’ psychosocial context before recommending individually tailored interventions for pregnant women.

### Strengths and Limitations

To our knowledge, our study is the first longitudinal study examining pregnant women’s pregnancy-induced symptoms from a wide range of distress follow-ups beginning early in the first trimester and ending at childbirth. We provide a clearer picture of the four patterns of pregnancy-induced symptom distress over the course of gestation, which could facilitate the identification of pregnant women who display particular trajectories of symptom distress and provide healthcare providers with new insights into symptom management. Clearly, it is important to assess pregnant women’s symptoms of distress properly to better understand relevant patterns. Healthcare providers could design intervention strategies for relieving or preventing pregnant women’s suffering and provide educational leaflets to promote pregnancy-induced symptom management during prenatal visits. Increased information and support from professionals could also help prevent or at least decrease the negative impact of symptom distress on the health of pregnant women and their fetuses.

This pilot study had a relatively small number (*n* = 95) of participants who were older and had a higher education level compared to Chen’s studies of 18,312 Taiwanese women [43]; therefore, we must be cautious in generalizing or extending our present results to a wider population. Taiwan is a moderate-to-high income country [44]. The prevalence of symptom distress is increasing among low-income groups [45,46]. We acknowledge that various patterns may exist, but more research is needed to understand the reasons behind a lack of “stable high” and “stable low” exhibiting among Taiwanese pregnant women. It could be related to a relatively small sample size, and further studies using empirical Bayes in HLM may deal with asymmetric time points of symptom distress. Therefore, future studies should use a larger group of participants or compare different contexts to establish a more solid consensus regarding the various trends or patterns in women’s pregnancy-induced symptom distress. Otherwise, our measurement tool had a good level validity and reliability, and it was already developed and examined on a small population of pregnant women in Taiwan [17,18]. Factor analysis methods can establish high-quality measures of those pregnancy-related symptom constructs with a large sample to construct their robust validity future.

## 5. Conclusions

Four patterns of distress symptoms were identified for pregnant women throughout their pregnancy. The most noticeable change pattern was the “Decreased then Increased” trend, which showed significantly lower symptom distress scores than the other patterns. Moreover, the point of transition to an increase in symptom distress scores was 22 weeks. The pregnant women in our sample generally tended to report distress levels that were below moderate (mean scores < 5). Physical symptoms were the main type of pregnancy-related distress symptoms for the pregnant women in this study; they included fatigue, lower back pain, abdominal distention and sleeping problems. Finally, the proper assessment of symptom distress continues to be important in order to have an early and clear understanding of their patterns of change, and to give providers clearer insights into symptom management and allow timely interventions. Such efforts could help prevent or decrease the negative impact of symptom distress on pregnant women and their fetuses.

## Figures and Tables

**Figure 1 ijerph-18-06333-f001:**
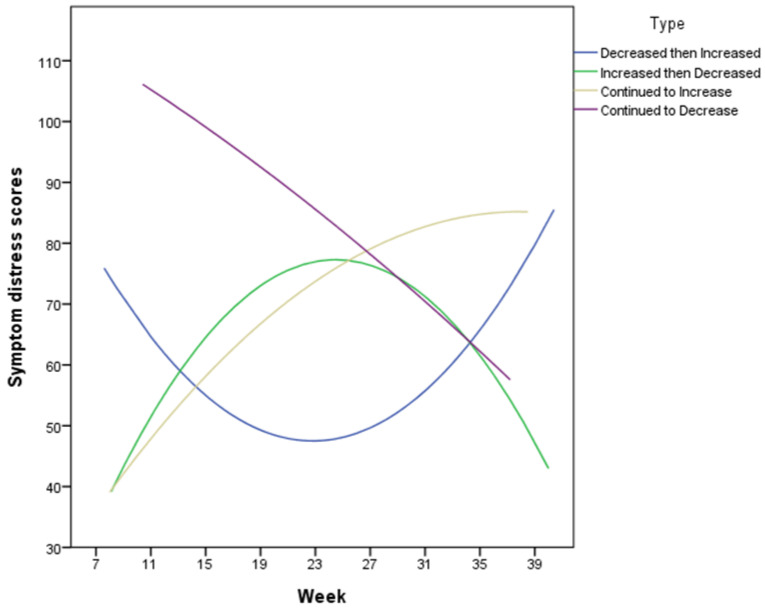
The pattern and changes in pregnancy-induced distress symptoms.

**Table 1 ijerph-18-06333-t001:** Participant characteristics (*n* = 95).

**Variables**	***n***	**%**
**Education level**		
Junior high school or below	8	8.4
College or university or above	87	91.6
**Family income**		
NT < 60,000	25	26.3
NT 60,000–100,000	54	56.9
NT > 100,000	16	16.8
**Occupation**		
Unemployed	13	13.7
Employed	82	86.3
**Parity**		
Nulliparous	52	54.7
Multiparous	43	45.3
**Abortion history**		
Yes	11	11.6
No	84	88.4
**Prepregnancy BMI**		
Underweight (<18.5)	9	9.5
Normal (18.5–24.9)	82	86.3
Overweight or obese (>25)	4	4.2
**Variables**	**Mean**	**SD**
Age	32.24	3.99
Gestational weight gain (kg)	9.11	4.22

**Table 2 ijerph-18-06333-t002:** Scores and transition time points for symptom distress for the 4 patterns (*n* = 95).

	Decreased then Increased (*n* = 54) a	Increased then Decreased (*n* = 10) b	Continuously Increased (*n* = 27) c	Continuously Decreased (*n* = 4) d	F	*p*	Post Hoc Test
	Mean (SE)	Mean (SE)	Mean (SE)	Mean (SE)			
Total	51.03(3.52)	63.35(7.63)	66.20(4.92)	79.41(12.83)	3.33 *	0.023	a < c a < d
Physical	41.42(2.65)	48.67(5.72)	53.37(3.69)	61.10(9.64)	3.20 *	0.027	a < c
Psychological	9.56(1.06)	14.68(2.29)	12.85(1.48)	18.22(2.85)	3.039 *	0.033	a < b a < d
Total average ^1^	2.32(0.16)	2.88(0.35)	3.02(0.22)	3.61(0.58)	3.313 *	0.024	a < c a < d
Turning point (weeks)			–	–	–	–	–
Total	22.02	26.77					
Physical	21.01	26.69					
Psychological	22.34	23.99					

Note: * *p* < 0.05; for the post hoc test in the column on the far right, a = Decreased then Increased; b = Increased then Decreased; c = Continuously Increased; d = Continuously Decreased. ^1^ Total average scores were the perceived level of distress on a 10-point scale (10 = extreme suffering; 0 = no suffering at all).

**Table 3 ijerph-18-06333-t003:** Distribution and scores of symptom distress for individual symptoms in the 4 patterns (*n* = 95).

	Decreased then Increased (*n* = 54) a	Increased then Decreased (*n* = 10) b	Continuously Increased (*n* = 27) c	Continuously Decreased (*n* = 4) d	F	*p*	Post Hoc Test
	Mean (SE)	Mean (SE)	Mean (SE)	Mean (SE)			
fatigue	4.06 (0.24)	4.59 (0.50)	5.22 (0.33)	6.23 (0.86)	4.14 **	0.008	a < c; a < d
2. frequent urination	3.84 (0.21)	4.26 (0.44)	4.76 (0.29)	4.32 (0.75)	2.26	0.086	
3. lower back pain	3.45 (0.26)	4.39 (0.56)	4.24 (0.37)	5.51 (0.96)	2.44	0.07	
4. abdominal distention	3.38 (0.25)	2.87 (0.54)	4.04 (0.35)	4.63 (0.93)	1.78	0.16	
5. constipation	3.19 (0.32)	3.04 (0.68)	4.3 (0.44)	2.97 (1.15)	1.63	0.19	
6. shortness of breath	2.70 (0.16)	3.13 (0.56)	3.67 (0.37)	4.07 (0.95)	1.91	0.13	
7. difficulty sleeping	2.54 (0.26)	3.58 (0.55)	3.99 (0.36)	5.42 (0.94)	5.78 **	0.001	a < c; a < d
8. swelling of legs	2.47 (0.24)	3.57 (0.51)	2.82 (0.33)	3.49 (0.87)	1.58	0.20	
9. nausea	2.28 (0.24)	1.70 (0.50)	2.38 (0.33)	3.06 (0.87)	0.74	0.53	
10. dizziness	2.10 (0.22)	2.34 (0.48)	2.57 (0.31)	2.41 (0.81)	0.53	0.67	
11. uterine contractions	1.85 (0.22)	2.86 (0.46)	2.76 (0.30)	3.45 (0.79)	3.45 *	0.02	a < b; a < c
12. leg cramps	1.84 (0.20)	2.78 (0.43)	2.40 (0.28)	3.15 (0.74)	2.36	0.077	
13. numbness/tingling in hands/feet	1.66 (0.22)	3.16 (0.47)	2.17 (0.31)	2.88 (0.81)	3.29 *	0.025	a < b
14. vomiting	1.66 (0.21)	1.47 (0.44)	1.67 (0.29)	2.50 (0.75)	0.48	0.70	
15. lumbopelvic pain	1.64(0.24)	2.39(0.50)	2.72(0.33)	2.53(0.86)	2.61	0.06	
16. hemorrhoids	1.46(0.24)	.98(0.51)	2.06(0.33)	1.63(0.87)	1.23	0.30	
17. diarrhea	1.14(0.18)	1.67(0.39)	1.83(0.26)	2.57(0.67)	2.73 *	0.049	a < c; a < d
worry	2.36(0.26)	3.07(0.56)	3.20(0.36)	4.65(0.94)	2.71	0.05	
2. anxiety	1.91(0.23)	3.14(0.49)	2.55(0.32)	3.97(0.82)	3.56 *	0.017	a < b; a < d
3. feeling upset	2.14(0.23)	3.09(0.50)	2.92(0.32)	4.33(0.85)	3.34 *	0.023	a < d
4. feeling unhappy	1.64(0.20)	2.78(0.44)	2.24(0.29)	3.01(0.74)	3.03 *	0.034	a < b
5. feeling depressed	1.49(0.20)	2.58(0.42)	1.95(0.27)	2.16(0.72)	2.12	0.10	

Note: * *p* < 0.05, ** *p* < 0.01; for the post hoc test in the column on the far right, a = Decreased then Increased; b = Increased then Decreased; c = Continuously Increased; d = Continuously Decreased.

## Data Availability

The data presented in this study are available on request from the corresponding author. The data are not publicly available due to funding restrictions and patient privacy.

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
