# Peer review of "Identifying Patterns of Symptom Distress in Pregnant Women: A Pilot Study"

_ijerph, 2021, doi:10.3390/ijerph18126333_

Round 1

Reviewer 1 Report

This is a well written manuscript dealing with an important issue faced by many pregnant women. This manuscript, which in my opinion should only be considered a pilot study because of its extremely small sample for the question being asked, identifies several potential trajectories of two types of symptom distress levels faced by women during pregnancy, physical and psychological, that can assist in providing treatment during pregnancy that can improve the women’s physical and mental health, potential reduce obstetric and birth complications, and subsequent long-term developmental issues in the exposed fetuses that are known to be associated with heightened levels of perinatal maternal distress. However, there are several issues with the manuscript that need to be addressed before it can be published.

Main concerns

  1. Sample Size determination
    1. Effect Size
      1. The authors indicate that the sample size was determined using G*Power analyses assuming a power of 80% and α = 0.05. However, they fail to provide the most important piece of the analysis: Effect Size. How was the ES calculated or estimated? What was the rationale for their choice of ES? This information is required.
    2. Why limit to 98 women
      1. Unless the purpose (which was not stated) of the present study was a pilot study, I question why the authors limited their sample to 98 women. A rationale needs to be provided explaining why the sample was not larger.
    3. Are all from same attending physician?
      1. I might be mistaken, but I feel that all women in the present study were obtained from the same attending physician. My concern stems from the phrase: ‘from an obstetrician in Taiwan’. Was this the case? If yes, this is very problematic as it restricts care to a single obstetrician which might have biased the findings. If no, please clarify how these women were recruited. From how many clinics/obstetricians, …
    4. Representative of Taiwanese national norms
      1. A brief comparison to the general Taiwanese population (particular against childbearing women) is necessary.
    5. Factor Analysis
      1. Are the authors certain that the physical and psychological factors assessed go together?
        1. Division has face value
        2. The authors calculated a Total distress score, and seemingly distress scores for Physical and Psychological distress levels. It also appears that the division between Physical and Psychological was arbitrary. While this division has strong face value, a factor analysis would have been a better approach to take, especially since the authors provided no rationale for examining the Physical and Psychological distress levels separately. Moreover, no rationale was provided for creating a Total Distress Score.
      2. Trajectories
        1. Only four patterns were observed. Interestingly, no women indicated Stable High or Stable Low. I am wondering if this is simply a product of the relatively small sample size for this type of pattern analysis. Moreover, could this also be a function of missing data points (please see my next concern)
      3. Quadratic Function
        1. How many participants provided only 2 measurements? The patterns reported by the authors take at least three data points to observe. It is important to indicate how many women provided only two data points. Moreover, do the women providing only two data points differ in any systematic manner relative to women providing three plus data points?
      4. What comes first? Physical or Psychological Distress
        1. Clearer indication of the association between physical and psychological factors is needed. What is the magnitude of these relationships?
        2. Did the women have similar total, physical, and psychological patterns. For example, what percentage of women were in the continuously decline pattern for ALL 3 measures?
        3. The authors mention times in pregnancy when patterns shifted (I am assuming this is for the total scores). Are the times in pregnancy the same for the physical and psychological patterns?
        4. What comes first, a change in physical distress levels or psychological distress levels?
          1. This is important for the development and implementation of interventions.
        5. Discussion
          1. Very small sample size to be making relatively strong statements about the observed patterns, particularly since no women apparently exhibited stable high or stable low distress levels.

Minor Concerns

  1. What is meant by: ‘Asian women tend to have significantly lower symptom distress than the general population, while Latina women tend to have significantly higher scores”. What is meant by ‘the general population”? What population? Global? Industrialized? Please clarify.

Author Response

Response to Reviewer 1 Comments

Thank you for your valuable time. On behalf of the authors, I have provided point-by-point explanations below to respond to your comments. 

Point 1: In general, the reviewer gave positive feedbacks, “This is a well written manuscript dealing with an important issue faced by many pregnant women.”

Point 2: The reviewer kindly provided his/her opinion of our study design, we have addressed their queries pointed out by reviewer 1, mainly in the title

Reviewer’s comments

Changes (in red) in the manuscript revision

This manuscript, which in my opinion should only be considered a pilot study because of its extremely small sample for the question being asked, identifies several potential trajectories of two types of symptom distress levels faced by women during pregnancy, physical and psychological, that can assist in providing treatment during pregnancy that can improve the women’s physical and mental health, potential reduce obstetric and birth complications, and subsequent long-term developmental issues in the exposed fetuses that are known to be associated with heightened levels of perinatal maternal distress.

The reviewer kindly provided his/her opinion of our study was considered a pilot study. We have addressed and change in the title and design subsection, as fellow:

1)    Line 3: We changed the title “Identifying patterns of symptom distress in pregnant women: A pilot study”

2)    Line 71: We have re-written the design subsection:” This pilot study used a longitudinal study design based on convenience sampling.”

Point 3: we have addressed reviewer 1’s comments, mainly in the sample size determination

Reviewer’s comments

Changes (in red) in the manuscript revision

1.    The authors indicate that the sample size was determined using G*Power analyses assuming a power of 80% and α = 0.05. However, they fail to provide the most important piece of the analysis: Effect Size. How was the ES calculated or estimated? What was the rationale for their choice of ES? This information is required.

Line 19:

In order to answer the question regarding the clarify the reviewer’s queries, we added a brief description to explain the effect size and sample size of our study, as follow:

1.    effect size = (µ1 − µ2)/σ = .2, .5, .8 for small, medium, and large effects (Cohen, 1988). We have set the medium effect size of 0.5 by Cohen’s analysis to clarify the difference between the time-varying. So we have revised our description and provided rationales for effect size in estimating the sample size.

Lines 88-92: “We used the G-Power software package (version 3.1) to calculate the necessary sample size, assuming a power of 80%, the medium effect size of 0.5 by Cohen’s analysis, and α = 0.05. The sample size estimate was caculated at 92 (N) in our study to reach satisfactory statistical power.  This number also provided us confidence since the rule of thumb in the longitudinal study design where the sample size at effect size 0.5 was 64 (for power =0.8 for a 2-tailed 0.05 test) (Hedeker, & Waternaux,1999).”

Reference list:

1.    Cohen, J. (1988). Statistical power analysis for the behavioral sciences (2nd ed.). Hillsdale, NJ: Lawrence Erlbaum Associates.

2.    Hedeker, Gibbons, & Waternaux (1999). Sample size estimation for longitudinal designs with attrition. Journal of Educational and Behavioral Statistics, 24:70-93.

Why limit to 98 women

1.    Unless the purpose (which was not stated) of the present study was a pilot study, I question why the authors limited their sample to 98 women. A rationale needs to be provided explaining why the sample was not larger.

A 'power analysis" is used to calculate the probability of finding an effect or a difference of a certain characteristic if it exists. The higher power, the lower risk of missing an actual effect (Serdar et al., 2021).

As we explained previously, a power analysis was conducted and it concluded that our current sample size is sufficient to not only address our study purpose but also explain the final outcomes.

Lines 89-90: The sample size estimate was calculated at 92 (N) in our study to reach satisfactory statistical power.

Reference:

Serdar, C. C., Cihan, M., Yücel, D., and Serdar, M. A. (2021). Sample size, power and effect size revisited: simplified and practical approaches in pre-clinical, clinical and laboratory studies. Biochem Med 15; 31(1): 010502. doi: 10.11613/BM.2021.010502

Are all from same attending physician?

1.    I might be mistaken, but I feel that all women in the present study were obtained from the same attending physician. My concern stems from the phrase: ‘from an obstetrician in Taiwan’. Was this the case? If yes, this is very problematic as it restricts care to a single obstetrician which might have biased the findings. If no, please clarify how these women were recruited. From how many clinics/obstetricians, …

Line 104:

Our participants were recruited from different attending physicians. There were more than 20 obstetricians participating in two teaching hospitals.

We made it clear in our original writing that “check-up from multiple obstetricians in Taiwan.”

Representative of Taiwanese national norms

1.    A brief comparison to the general Taiwanese population (particular against childbearing women) is necessary.

Line 73-74:

In Taiwan, only a small portion of pregnant women (0.08%) gave birth by midwives (Health Promotion Administration; Ministry of Health and Welfare, 2021). Most Taiwanese pregnant women believed that childbirth and delivery was much safer under obstetricians’ supervision than delivered by midwives at hospital in Taiwan.

We have supplied information about the Taiwanese national norms in the text:

Line 74-75: According to the Taiwanese Health Promotion Administration report of Ministry of Health and Welfare, “”more than 99.9% of women, regardless of their gestation statuses or pregnancy related risks, give birth by obstetricians at hospitals in Taiwan.

Reference:

16.Health Promotion Administration; Ministry of Health and Welfare (January 29, 2021) https://www.gender.ey.gov.tw/gecdb/Stat_Statistics_DetailData.aspx?sn=UNgbUqD746EaB%2fcBxzuqGw%3d%3d&d=194q2o4%2botzoYO%2b8OAMYew%3d%3d”

Point 4: We have addressed reviewer 1’s comments in the factor analysis section.

Reviewer’s comments

Changes (in red) in the manuscript revision

1. Are the authors certain that the physical and psychological factors assessed go together?

1.1  Division has face value

1.2. The authors calculated a Total distress score, and seemingly distress scores for Physical and Psychological distress levels. It also appears that the division between Physical and Psychological was arbitrary. While this division has strong face value, a factor analysis would have been a better approach to take, especially since the authors provided no rationale for examining the Physical and Psychological distress levels separately. Moreover, no rationale was provided for creating a Total Distress Score.

1.    Yes, we were used 22-item questionnaire to assess both physical and psychological distress together and calculate their scores of total distresses. It already presented on lines 125-126.

2.    Line 129: We added the CVI values of distress scores in total, physical and psychological aspects: “The CVI values were 96.25, 97.5% and 95%, respectively. “

3.    Moreover, we have added the limitation about the validity of our questionnaire

Line 368-370: “Otherwise, our measurement tool was already developed and examined on small population of pregnant women in Taiwan [17-18]. A factor analysis approach for a large sample to construct their robust validity future.”

4.    We have made changes to explain the total scores in our study.

Line 125-126: “In aspects of physical distress and psychological distress mean total distress, the highest possible total score was 220.” 

2.Trajectories

2.1. Only four patterns were observed. Interestingly, no women indicated Stable High or Stable Low. I am wondering if this is simply a product of the relatively small sample size for this type of pattern analysis. Moreover, could this also be a function of missing data points (please see my next concern)

1.    For all participants in our study, the patterns were identified, hence the estimation of our trajectory parameters, by quadratic growth model. The categories of our trajectory parameter were estimated according to our linear and quadratic analyses.

While the trajectory parameters of linear and quadratic for the pattern of “stable HIGH” was 0 and insignificantly with significantly large intercepts. On the other hand, the trajectory parameters of linear and quadratic for the pattern of “stable LOW” was also 0 and insignificantly, yet their intercepts were insignificant or small. None of our participants have exhibited these two patterns (stable high or low).

2.  We may also estimate the growth model by HLM. The quadratic parameter representing for the growth curve could be estimated by more than 3 data points. Even so, when participants were examined fewer than three times (limited data points), empirical Bayes may be used in HLM to estimate each individual’s trajectory.

Therefore, our sample size was sufficient according to the power analysis, but further studies may be necessary to examine the relationship between sample size and these two patterns. With a larger sample size, pregnant women may show various patterns. Another statistical analysis such as empirical Bayes in HLM may also provide answers to this.

3. Quadratic Function

3.1. How many participants provided only 2 measurements? The patterns reported by the authors take at least three data points to observe. It is important to indicate how many women provided only two data points. Moreover, do the women providing only two data points differ in any systematic manner relative to women providing three plus data points?

1.    Line 96: We wrote that “Only 5.3%(n=5) participants completed two measurements.”  

2.    The longitudinal study need collect least two data points for knowing their changing in quadratic trajectory. If we consistently collected only two data points, the individual trajectory will be estimated by empirical Bayes of HLM; if we collected three data points, the quadratic trajectory could be examined. HLM allows asymmetric data whereas participants provide either two or more than tree time points; their changing trend may be estimated in a longitudinal study design.

3.    Line 132-134: We estimated the trend in the distress scores by hierarchical linear modelling (HLM). Because each participant had asymmetric time points of symptom distress scores in the data-collecting period, the HLM permits the repeated measures within a nested structure [19].  

4.    What comes first? Physical or Psychological Distress

4.1. Clearer indication of the association between physical and psychological factors is needed. What is the magnitude of these relationships?

4.2. Did the women have similar total, physical, and psychological patterns. For example, what percentage of women were in the continuously decline pattern for ALL 3 measures?

4.3. The authors mention times in pregnancy when patterns shifted (I am assuming this is for the total scores). Are the times in pregnancy the same for the physical and psychological patterns?

4.4  What comes first, a change in physical distress levels or psychological distress levels?

4.4.1.     This is important for the development and implementation of interventions.

1.    The distress level is higher and significantly correlation between physical and psychological (r=0.73, p<.05). pregnant women had higher physical distress level would be with higher psychological distress level. However,

2.    Our participants had the similar distress pattern. We have list percentage of the all pattern in distress of total, physical and psychological, see the below table:

Decreased then Increased

Increased then Decreased

Continuously Increased

Continuously Decreased

n(%)

n(%)

n(%)

n(%)

Total

54(56.8)

27(28.4)

10(10.5)

4(4.2)

physical

45(47.4)

35(36.8)

12(12.6)

3(3.2)

psychological

61(64.2)

21(22.1)

7(7.4)

6(6.3)

3.    We presented the turning point in week is total scores. In order to answer the question regarding the clarify the reviewer’s queries, we added a brief description to explain the turning points of physical and psychological distress, as follow:

1)    Table 2: of physical and psychological distress on the column of “Decreased then increase” and “Increased then decreased”, see (page 7)

2)    Line 201-203: ” From the most noticeable changes pattern of  “Decrease then increase “ and the turn point of increasing of total, physical and psychological aspects was at 22.02, 21.01 and 22.34 week, respectively. “

3)  A brief description has been added to explain the physical distress comes early than psychological distress.

Line 260-262:We also found that physical symptom distress increased first (21.01 weeks) than psychological distress (22.34 weeks) under the most noticeable changing pattern of the “Decreased then Increased”.”

4.    This important finding was added in discussion section, as follow:”

Line 284-286: “Furthermore, self-care related education about the ways to relieve early physical symptom distress should be provided soon to pregnant women at the first prenatal care visit. “

Point 4: We have addressed reviewer 1’s comments in the discussion.

Reviewer’s comments

Changes (in red) in the manuscript revision

Discussion

1.    Very small sample size to be making relatively strong statements about the observed patterns, particularly since no women apparently exhibited stable high or stable low distress levels.

A brief description has been added to explain our sample size on discussion section, as follow:

1)       Line 358: “This pilot study was had a relatively small number (n=95)….”

2)       Line 361-365: “We acknowledge that various patterns may exist, but more research is needed to understand the reasons behind a lack of “stable high” and “stable low” exhibiting among Taiwanese pregnant women. It could be related to a relatively small sample size, and further studies using empirical Bayes in HLM may deal with asymmetric time points of symptom distress”.

Reviewer’s comments

Changes (in red) in the manuscript revision

What is meant by: ‘Asian women tend to have significantly lower symptom distress than the general population, while Latina women tend to have significantly higher scores”. What is meant by ‘the general population”? What population? Global? Industrialized? Please clarify.

We have re-written our text.

Line 333-335: ” Asian women tend to have significantly lower symptom distress than the general populations of Euro-Americans, while the Latina women tend to have significantly higher [1].”

Reviewer 2 Report

Dear authors,

The first sentence of the abstract establish a causal relationship without any logic. I would change it "The enlarged uterus and developing fetus lead to symptom distress during pregnancy".

This sentence makes no sense to me and I think it will be hard to understand for the readers. "The most noticeable changes in symptom distress scores 21 occurred for the “Decreased then Increased” (56.8%) pattern, followed by the “Continuously 22 Increased” (28.4%), “Increased then Decreased” (10.5%) and “Continuously Decreased” (4.2%) 23 patterns. In the “Decreased then Increased”. It is only specified in the methods and Figure 1 should be in the methods because it shows the meaning.

Which week is this one? "22.02 weeks" If this value is a mean you should specify it.

I do not see a clear definition of what the authors mean when they talk about "symptom distress" in the introduction.

Who collected the data? Physicians? Nurses?

An "ethical aspects" subsection must be included in the methods.

The limitations must included that you have used an own and no validated questionnaire.

Kind regards

Author Response

Response to Reviewer 2 Comments

Thank you for your valuable time. On behalf of the authors, I have provided point-by-point explanations below to respond to your comments. 

 Point 1: We have addressed queries pointed out by reviewer 2, mainly in the abstract.

Reviewer’s comments

Changes (in red) in the manuscript revision

The first sentence of the abstract establish a causal relationship without any logic. I would change it "The enlarged uterus and developing fetus lead to symptom distress during pregnancy".

Thank you for the comment. We have re-written the text:

Line 16-18: “During pregnancy, women’s enlarged uterus and developing fetus lead to symptom distress; in turn, physical and psychological aspects of symptom distress is often associated with adverse prenatal and birth outcomes. “

This sentence makes no sense to me and I think it will be hard to understand for the readers. "The most noticeable changes in symptom distress scores  occurred for the “Decreased then Increased” (56.8%) pattern, followed by the “Continuously  Increased” (28.4%), “Increased then Decreased” (10.5%) and “Continuously Decreased” (4.2%)  patterns. In the “Decreased then Increased”. It is only specified in the methods and Figure 1 should be in the methods because it shows the meaning.

Thank you for the comment. We have re-written the text:

Line 22-28: “More than half of the pregnant women’ experienced symptom distress manifested in a pattern depicted to be “Decreased then Increased” (56.8%).Other noticeable patterns were “Continuously Increased” (28.4%), “Increased then Decreased” (10.5%) and “Continuously Decreased” (4.2%), respectively. “

Which week is this one? "22.02 weeks" If this value is a mean you should specify it.

We have re-written the text to make it clear:

Line 26-27: “It is worth noting that most pregnant women transited and increased their symptom distress, revealed by their total scores, at the second trimester (mean at 22.02 weeks) of pregnancy.”

  1. Point 2 Introduction:

I do not see a clear definition of what the authors mean when they talk about "symptom distress" in the introduction.

We have re-written to make it clear.

Line 52-53: “Symptom distress is an individual’s perceived frequency of particular symptoms, as well as the severity or the level of distress caused by the symptoms [13].”

Point 2 Materials and Methods: Thank you for recognizing our work.

Who collected the data? Physicians? Nurses?

Both nurses and physician are researchers to recruit participants and collect data, physicians confirmed participants’ gestation weeks and health status during pregnancy.

We have revised the description, as follow:

Line 98-102: “Symptom-related measurement began during 8-12 weeks of the first trimester after the pregnancy was confirmed by obstetricians, and at least two subsequent measurements were collected by researchers every 4 weeks in the second trimester and every two weeks in the third trimester during prenatal visits until childbirth.”

An "ethical aspects" subsection must be included in the methods.

Line 162-165:  

We have added the subsection of ethical in the method section, as follow: “

2.6 Ethical considerations

 This study was approved by the Institutional Review Board of Mackay Medical Hospital in Taipei, Taiwan (#16MMHIS135). The consent forms signed by the participants included both a brief description of the study and an explicit assurance of confidentiality.”

  1. Point 4 Limitation:

The limitations must include that you have used an own and no validated questionnaire.

Line 368-370:

We have addressed the limitation as follow” Otherwise, our measurement tool was already developed and examined on small population of pregnant women in Taiwan [17-18]. A factor analysis approach for a large sample to construct their robust validity future.  “

Response to Reviewer 2 Comments

Thank you for your valuable time. On behalf of the authors, I have provided point-by-point explanations below to respond to your comments. 

 Point 1: We have addressed queries pointed out by reviewer 2, mainly in the abstract.

Reviewer’s comments

Changes (in red) in the manuscript revision

The first sentence of the abstract establish a causal relationship without any logic. I would change it "The enlarged uterus and developing fetus lead to symptom distress during pregnancy".

Thank you for the comment. We have re-written the text:

Line 16-18: “During pregnancy, women’s enlarged uterus and developing fetus lead to symptom distress; in turn, physical and psychological aspects of symptom distress is often associated with adverse prenatal and birth outcomes. “

This sentence makes no sense to me and I think it will be hard to understand for the readers. "The most noticeable changes in symptom distress scores  occurred for the “Decreased then Increased” (56.8%) pattern, followed by the “Continuously  Increased” (28.4%), “Increased then Decreased” (10.5%) and “Continuously Decreased” (4.2%)  patterns. In the “Decreased then Increased”. It is only specified in the methods and Figure 1 should be in the methods because it shows the meaning.

Thank you for the comment. We have re-written the text:

Line 22-28: “More than half of the pregnant women’ experienced symptom distress manifested in a pattern depicted to be “Decreased then Increased” (56.8%).Other noticeable patterns were “Continuously Increased” (28.4%), “Increased then Decreased” (10.5%) and “Continuously Decreased” (4.2%), respectively. “

Which week is this one? "22.02 weeks" If this value is a mean you should specify it.

We have re-written the text to make it clear:

Line 26-27: “It is worth noting that most pregnant women transited and increased their symptom distress, revealed by their total scores, at the second trimester (mean at 22.02 weeks) of pregnancy.”

  1. Point 2 Introduction:

I do not see a clear definition of what the authors mean when they talk about "symptom distress" in the introduction.

We have re-written to make it clear.

Line 52-53: “Symptom distress is an individual’s perceived frequency of particular symptoms, as well as the severity or the level of distress caused by the symptoms [13].”

Point 2 Materials and Methods: Thank you for recognizing our work.

Who collected the data? Physicians? Nurses?

Both nurses and physician are researchers to recruit participants and collect data, physicians confirmed participants’ gestation weeks and health status during pregnancy.

We have revised the description, as follow:

Line 98-102: “Symptom-related measurement began during 8-12 weeks of the first trimester after the pregnancy was confirmed by obstetricians, and at least two subsequent measurements were collected by researchers every 4 weeks in the second trimester and every two weeks in the third trimester during prenatal visits until childbirth.”

An "ethical aspects" subsection must be included in the methods.

Line 162-165:  

We have added the subsection of ethical in the method section, as follow: “

2.6 Ethical considerations

 This study was approved by the Institutional Review Board of Mackay Medical Hospital in Taipei, Taiwan (#16MMHIS135). The consent forms signed by the participants included both a brief description of the study and an explicit assurance of confidentiality.”

  1. Point 4 Limitation:

The limitations must include that you have used an own and no validated questionnaire.

Line 368-370:

We have addressed the limitation as follow” Otherwise, our measurement tool was already developed and examined on small population of pregnant women in Taiwan [17-18]. A factor analysis approach for a large sample to construct their robust validity future.  “

Response to Reviewer 2 Comments

Thank you for your valuable time. On behalf of the authors, I have provided point-by-point explanations below to respond to your comments. 

 Point 1: We have addressed queries pointed out by reviewer 2, mainly in the abstract.

Reviewer’s comments

Changes (in red) in the manuscript revision

The first sentence of the abstract establish a causal relationship without any logic. I would change it "The enlarged uterus and developing fetus lead to symptom distress during pregnancy".

Thank you for the comment. We have re-written the text:

Line 16-18: “During pregnancy, women’s enlarged uterus and developing fetus lead to symptom distress; in turn, physical and psychological aspects of symptom distress is often associated with adverse prenatal and birth outcomes. “

This sentence makes no sense to me and I think it will be hard to understand for the readers. "The most noticeable changes in symptom distress scores  occurred for the “Decreased then Increased” (56.8%) pattern, followed by the “Continuously  Increased” (28.4%), “Increased then Decreased” (10.5%) and “Continuously Decreased” (4.2%)  patterns. In the “Decreased then Increased”. It is only specified in the methods and Figure 1 should be in the methods because it shows the meaning.

Thank you for the comment. We have re-written the text:

Line 22-28: “More than half of the pregnant women’ experienced symptom distress manifested in a pattern depicted to be “Decreased then Increased” (56.8%).Other noticeable patterns were “Continuously Increased” (28.4%), “Increased then Decreased” (10.5%) and “Continuously Decreased” (4.2%), respectively. “

Which week is this one? "22.02 weeks" If this value is a mean you should specify it.

We have re-written the text to make it clear:

Line 26-27: “It is worth noting that most pregnant women transited and increased their symptom distress, revealed by their total scores, at the second trimester (mean at 22.02 weeks) of pregnancy.”

  1. Point 2 Introduction:

I do not see a clear definition of what the authors mean when they talk about "symptom distress" in the introduction.

We have re-written to make it clear.

Line 52-53: “Symptom distress is an individual’s perceived frequency of particular symptoms, as well as the severity or the level of distress caused by the symptoms [13].”

Point 2 Materials and Methods: Thank you for recognizing our work.

Who collected the data? Physicians? Nurses?

Both nurses and physician are researchers to recruit participants and collect data, physicians confirmed participants’ gestation weeks and health status during pregnancy.

We have revised the description, as follow:

Line 98-102: “Symptom-related measurement began during 8-12 weeks of the first trimester after the pregnancy was confirmed by obstetricians, and at least two subsequent measurements were collected by researchers every 4 weeks in the second trimester and every two weeks in the third trimester during prenatal visits until childbirth.”

An "ethical aspects" subsection must be included in the methods.

Line 162-165:  

We have added the subsection of ethical in the method section, as follow: “

2.6 Ethical considerations

 This study was approved by the Institutional Review Board of Mackay Medical Hospital in Taipei, Taiwan (#16MMHIS135). The consent forms signed by the participants included both a brief description of the study and an explicit assurance of confidentiality.”

  1. Point 4 Limitation:

The limitations must include that you have used an own and no validated questionnaire.

Line 368-370:

We have addressed the limitation as follow” Otherwise, our measurement tool was already developed and examined on small population of pregnant women in Taiwan [17-18]. A factor analysis approach for a large sample to construct their robust validity future.  “

Round 2

Reviewer 1 Report

Overall, the authors did an excellent job in address my concerns. However, there are a few issues that need to be clarified.

1. Hospitals

In their response to this issue the authors indicate that the sample was drawn from 2 teaching hospitals. This information should be clearly stated in the Methods. Moreover, in their response they indicate that the sample was drawn from more than 20 obstetricians from these hospitals. It would be best if the specific number of obstetricians is clearly stated.

2. A brief comparison to the general Taiwanese population (particular against childbearing women) is necessary

I think there might have been some confusion about what I was asking. My question was whether the women who participated in this study differed from the general population in any significant manner. For example, were the women in this sample better educated, older/younger, had previous pregnancies, ... than the general population. Knowing whether the sample is representative of the general population is vital before general statements about the women's physical and psychological experiences can be drawn. This is particularly important since the sample was obtained from two teaching hospital, which I am assuming are located in urban and not rural areas. Please describe how this study sample is similar to and different from the general childbearing population of Taiwan.

From the most noticeable changes pattern of “Decrease then increase “ and the turn point of increasing of total, physical and psychological aspects 201 was at 22.02, 21.01and 22.34 week, respectively (Table 2).

This sentence is difficult to interpret. And it only deals with one pattern of change. Should the sentence read:

For the Decrease then increase trajectory, while the overall pattern changed at 22.02 weeks, physical distress levels increased earlier (21.01 weeks) relative to psychological distress levels (22.34 weeks).

Please provide a better explanation for what the authors think this means. Do physical factors drive the psychological distress level changes. If yes, why should we be concerned about this pattern of change.

Is the same pattern (physical before psychological) seen in the Increase then Decrease group? If yes, what does this mean. If no, why does it mean?

Otherwise, our measurement tool was already developed and examined on small population of pregnant woomen in Taiwan [17-18]. A factor analysis approach for a large sample to construct their robust validity future.

The cited references are not appropriate as the first sentence above is referring to previous research using the assessment tool. Please provide the appropriate reference(s).

Women (woomen) is misspelled

Finally, it is unclear what the authors are trying to communicate with the second sentence. Please clarify

Author Response

Response to the Reviewer’s Comments

Thank you for your valuable time. On behalf of the authors, I have provided point-by-point explanations below to respond to your comments. 

Point 1: In general, the reviewer gave positive feedbacks, “Overall, the authors did an excellent job in address my concerns.”

Point 2: The reviewer kindly provided his/her opinion of specifying the number of obstetricians and participating hospitals in methods section.

Reviewer’s comments

Changes (in red) in the manuscript revision

1. Hospitals

In their response to this issue the authors indicate that the sample was drawn from 2 teaching hospitals. This information should be clearly stated in the Methods. Moreover, in their response they indicate that the sample was drawn from more than 20 obstetricians from these hospitals. It would be best if the specific number of obstetricians is clearly stated.

We have re-written relative texts in the abstract and method section, as fellow:

1)    Line 20:

“…, from the prenatal wards of two teaching hospitals in northern Taiwan.”

2)    Line 82-83:

” …,from the prenatal outpatient departments with a total of 28 obstetricians of two teaching hospitals in Taipei, Taiwan.”

Point 3: we have addressed the reviewer’s comments, mainly in the comparison to the general Taiwanese population

Reviewer’s comments

Changes (in red) in the manuscript revision

2. A brief comparison to the general Taiwanese population (particular against childbearing women) is necessary

I think there might have been some confusion about what I was asking. My question was whether the women who participated in this study differed from the general population in any significant manner. For example, were the women in this sample better educated, older/younger, had previous pregnancies, ... than the general population. Knowing whether the sample is representative of the general population is vital before general statements about the women's physical and psychological experiences can be drawn.

This is particularly important since the sample was obtained from two teaching hospital, which I am assuming are located in urban and not rural areas. Please describe how this study sample is similar to and different from the general childbearing population of Taiwan.

Line 365-368:

In order to answer the question regarding the clarify the reviewer’s queries, we added a brief comparison our sample and general Taiwanese population, as follow:

“Since our sample were mostly employed, they were relatively older, higher educated with better family income than the general population, compared to the 18,312 Taiwanese women included in the recent Taiwan Birth Cohort Study (TBCS) data bank (Chen, 2020).”

Reference:

Chen, C.-N.; Yu, H.-C.; Chou, A.-K. Association between Maternal Pre-pregnancy Body Mass Index and Breastfeeding Duration in Taiwan: A Population-Based Cohort Study. Nutrients 2020, 12, 2361. https://doi.org/10.3390/nu12082361

Point 4: We have addressed the reviewer’s comments in the result section.

Reviewer’s comments

Changes (in red) in the manuscript revision

From the most noticeable changes pattern of “Decrease then increase “ and the turn point of increasing of total, physical and psychological aspects 201 was at 22.02, 21.01and 22.34 week, respectively (Table 2).

This sentence is difficult to interpret. And it only deals with one pattern of change. Should the sentence read:

For the Decrease then increase trajectory, while the overall pattern changed at 22.02 weeks, physical distress levels increased earlier (21.01 weeks) relative to psychological distress levels (22.34 weeks).

We have re-written our text, according reviewer’s comments.

LINE:201-203

“For the Decrease then increase trajectory, while the overall pattern changed at 22.02 weeks, physical distress levels increased earlier (21.01 weeks) relative to psychological distress levels (22.34 weeks).” 

Point 5: We have addressed the reviewer’s comments in the discussion section.

Reviewer’s comments

Changes (in red) in the manuscript revision

Please provide a better explanation for what the authors think this means. Do physical factors drive the psychological distress level changes. If yes, why should we be concerned about this pattern of change.

Is the same pattern (physical before psychological) seen in the Increase then Decrease group? If yes, what does this mean. If no, why does it mean?

In order to answer the question, we added a brief description in the discussion section as follow:

1). Line 265-271:

“We also found that physical symptom distress changed prior to the psychological distress. This pattern has signified the importance that health care providers may prevent both types of symptom distress by first relieving the physical symptoms. In particular, education on self-care could be provided emphasized on alleviating physical symptoms in the first trimester. Early reduction of physical symptom distress would be key to help with increased levels of perceived psychological and overall symptom distress”.

2). Line 239-242:

“The key cause of increased physical, and later psychological, symptom distress may be originated from maternal physiological changes of fetal growth and increasing uterine size at 21-24 gestational weeks [24,25]. ……We believe that the anatomical and physical changes have resulted in increased distress levels along with gestation weeks. Such changes not only affect physical but psychological and overall symptom distress. “

Point 6: We have addressed reviewer 1’s comments in the discussion.

Reviewer’s comments

Changes (in red) in the manuscript revision

Otherwise, our measurement tool was already developed and examined on small population of pregnant woomen in Taiwan [17-18]. A factor analysis approach for a large sample to construct their robust validity future.

The cited references are not appropriate as the first sentence above is referring to previous research using the assessment tool. Please provide the appropriate reference(s).

Women (woomen) is misspelled

Finally, it is unclear what the authors are trying to communicate with the second sentence. Please clarify

Line: 378-382

We have re-written our text.

“Although we recommend the convenient measurement tool of fair psychometrics in this study, developed and examined among pregnant women in Taiwan [17-18], a further robust examination is required such as factor analyses for constructing the validity of pregnant women’s symptom distress in a larger, cross-cultural sample.”

Response to the Reviewer’s Comments

Thank you for your valuable time. On behalf of the authors, I have provided point-by-point explanations below to respond to your comments. 

Point 1: In general, the reviewer gave positive feedbacks, “Overall, the authors did an excellent job in address my concerns.”

Point 2: The reviewer kindly provided his/her opinion of specifying the number of obstetricians and participating hospitals in methods section.

Reviewer’s comments

Changes (in red) in the manuscript revision

1. Hospitals

In their response to this issue the authors indicate that the sample was drawn from 2 teaching hospitals. This information should be clearly stated in the Methods. Moreover, in their response they indicate that the sample was drawn from more than 20 obstetricians from these hospitals. It would be best if the specific number of obstetricians is clearly stated.

We have re-written relative texts in the abstract and method section, as fellow:

1)    Line 20:

“…, from the prenatal wards of two teaching hospitals in northern Taiwan.”

2)    Line 82-83:

” …,from the prenatal outpatient departments with a total of 28 obstetricians of two teaching hospitals in Taipei, Taiwan.”

Point 3: we have addressed the reviewer’s comments, mainly in the comparison to the general Taiwanese population

Reviewer’s comments

Changes (in red) in the manuscript revision

2. A brief comparison to the general Taiwanese population (particular against childbearing women) is necessary

I think there might have been some confusion about what I was asking. My question was whether the women who participated in this study differed from the general population in any significant manner. For example, were the women in this sample better educated, older/younger, had previous pregnancies, ... than the general population. Knowing whether the sample is representative of the general population is vital before general statements about the women's physical and psychological experiences can be drawn.

This is particularly important since the sample was obtained from two teaching hospital, which I am assuming are located in urban and not rural areas. Please describe how this study sample is similar to and different from the general childbearing population of Taiwan.

Line 365-368:

In order to answer the question regarding the clarify the reviewer’s queries, we added a brief comparison our sample and general Taiwanese population, as follow:

“Since our sample were mostly employed, they were relatively older, higher educated with better family income than the general population, compared to the 18,312 Taiwanese women included in the recent Taiwan Birth Cohort Study (TBCS) data bank (Chen, 2020).”

Reference:

Chen, C.-N.; Yu, H.-C.; Chou, A.-K. Association between Maternal Pre-pregnancy Body Mass Index and Breastfeeding Duration in Taiwan: A Population-Based Cohort Study. Nutrients 2020, 12, 2361. https://doi.org/10.3390/nu12082361

Point 4: We have addressed the reviewer’s comments in the result section.

Reviewer’s comments

Changes (in red) in the manuscript revision

From the most noticeable changes pattern of “Decrease then increase “ and the turn point of increasing of total, physical and psychological aspects 201 was at 22.02, 21.01and 22.34 week, respectively (Table 2).

This sentence is difficult to interpret. And it only deals with one pattern of change. Should the sentence read:

For the Decrease then increase trajectory, while the overall pattern changed at 22.02 weeks, physical distress levels increased earlier (21.01 weeks) relative to psychological distress levels (22.34 weeks).

We have re-written our text, according reviewer’s comments.

LINE:201-203

“For the Decrease then increase trajectory, while the overall pattern changed at 22.02 weeks, physical distress levels increased earlier (21.01 weeks) relative to psychological distress levels (22.34 weeks).” 

Point 5: We have addressed the reviewer’s comments in the discussion section.

Reviewer’s comments

Changes (in red) in the manuscript revision

Please provide a better explanation for what the authors think this means. Do physical factors drive the psychological distress level changes. If yes, why should we be concerned about this pattern of change.

Is the same pattern (physical before psychological) seen in the Increase then Decrease group? If yes, what does this mean. If no, why does it mean?

In order to answer the question, we added a brief description in the discussion section as follow:

1). Line 265-271:

“We also found that physical symptom distress changed prior to the psychological distress. This pattern has signified the importance that health care providers may prevent both types of symptom distress by first relieving the physical symptoms. In particular, education on self-care could be provided emphasized on alleviating physical symptoms in the first trimester. Early reduction of physical symptom distress would be key to help with increased levels of perceived psychological and overall symptom distress”.

2). Line 239-242:

“The key cause of increased physical, and later psychological, symptom distress may be originated from maternal physiological changes of fetal growth and increasing uterine size at 21-24 gestational weeks [24,25]. ……We believe that the anatomical and physical changes have resulted in increased distress levels along with gestation weeks. Such changes not only affect physical but psychological and overall symptom distress. “

Point 6: We have addressed reviewer 1’s comments in the discussion.

Reviewer’s comments

Changes (in red) in the manuscript revision

Otherwise, our measurement tool was already developed and examined on small population of pregnant woomen in Taiwan [17-18]. A factor analysis approach for a large sample to construct their robust validity future.

The cited references are not appropriate as the first sentence above is referring to previous research using the assessment tool. Please provide the appropriate reference(s).

Women (woomen) is misspelled

Finally, it is unclear what the authors are trying to communicate with the second sentence. Please clarify

Line: 378-382

We have re-written our text.

“Although we recommend the convenient measurement tool of fair psychometrics in this study, developed and examined among pregnant women in Taiwan [17-18], a further robust examination is required such as factor analyses for constructing the validity of pregnant women’s symptom distress in a larger, cross-cultural sample.”

Response to the Reviewer’s Comments

Thank you for your valuable time. On behalf of the authors, I have provided point-by-point explanations below to respond to your comments. 

Point 1: In general, the reviewer gave positive feedbacks, “Overall, the authors did an excellent job in address my concerns.”

Point 2: The reviewer kindly provided his/her opinion of specifying the number of obstetricians and participating hospitals in methods section.

Reviewer’s comments

Changes (in red) in the manuscript revision

1. Hospitals

In their response to this issue the authors indicate that the sample was drawn from 2 teaching hospitals. This information should be clearly stated in the Methods. Moreover, in their response they indicate that the sample was drawn from more than 20 obstetricians from these hospitals. It would be best if the specific number of obstetricians is clearly stated.

We have re-written relative texts in the abstract and method section, as fellow:

1)    Line 20:

“…, from the prenatal wards of two teaching hospitals in northern Taiwan.”

2)    Line 82-83:

” …,from the prenatal outpatient departments with a total of 28 obstetricians of two teaching hospitals in Taipei, Taiwan.”

Point 3: we have addressed the reviewer’s comments, mainly in the comparison to the general Taiwanese population

Reviewer’s comments

Changes (in red) in the manuscript revision

2. A brief comparison to the general Taiwanese population (particular against childbearing women) is necessary

I think there might have been some confusion about what I was asking. My question was whether the women who participated in this study differed from the general population in any significant manner. For example, were the women in this sample better educated, older/younger, had previous pregnancies, ... than the general population. Knowing whether the sample is representative of the general population is vital before general statements about the women's physical and psychological experiences can be drawn.

This is particularly important since the sample was obtained from two teaching hospital, which I am assuming are located in urban and not rural areas. Please describe how this study sample is similar to and different from the general childbearing population of Taiwan.

Line 365-368:

In order to answer the question regarding the clarify the reviewer’s queries, we added a brief comparison our sample and general Taiwanese population, as follow:

“Since our sample were mostly employed, they were relatively older, higher educated with better family income than the general population, compared to the 18,312 Taiwanese women included in the recent Taiwan Birth Cohort Study (TBCS) data bank (Chen, 2020).”

Reference:

Chen, C.-N.; Yu, H.-C.; Chou, A.-K. Association between Maternal Pre-pregnancy Body Mass Index and Breastfeeding Duration in Taiwan: A Population-Based Cohort Study. Nutrients 2020, 12, 2361. https://doi.org/10.3390/nu12082361

Point 4: We have addressed the reviewer’s comments in the result section.

Reviewer’s comments

Changes (in red) in the manuscript revision

From the most noticeable changes pattern of “Decrease then increase “ and the turn point of increasing of total, physical and psychological aspects 201 was at 22.02, 21.01and 22.34 week, respectively (Table 2).

This sentence is difficult to interpret. And it only deals with one pattern of change. Should the sentence read:

For the Decrease then increase trajectory, while the overall pattern changed at 22.02 weeks, physical distress levels increased earlier (21.01 weeks) relative to psychological distress levels (22.34 weeks).

We have re-written our text, according reviewer’s comments.

LINE:201-203

“For the Decrease then increase trajectory, while the overall pattern changed at 22.02 weeks, physical distress levels increased earlier (21.01 weeks) relative to psychological distress levels (22.34 weeks).” 

Point 5: We have addressed the reviewer’s comments in the discussion section.

Reviewer’s comments

Changes (in red) in the manuscript revision

Please provide a better explanation for what the authors think this means. Do physical factors drive the psychological distress level changes. If yes, why should we be concerned about this pattern of change.

Is the same pattern (physical before psychological) seen in the Increase then Decrease group? If yes, what does this mean. If no, why does it mean?

In order to answer the question, we added a brief description in the discussion section as follow:

1). Line 265-271:

“We also found that physical symptom distress changed prior to the psychological distress. This pattern has signified the importance that health care providers may prevent both types of symptom distress by first relieving the physical symptoms. In particular, education on self-care could be provided emphasized on alleviating physical symptoms in the first trimester. Early reduction of physical symptom distress would be key to help with increased levels of perceived psychological and overall symptom distress”.

2). Line 239-242:

“The key cause of increased physical, and later psychological, symptom distress may be originated from maternal physiological changes of fetal growth and increasing uterine size at 21-24 gestational weeks [24,25]. ……We believe that the anatomical and physical changes have resulted in increased distress levels along with gestation weeks. Such changes not only affect physical but psychological and overall symptom distress. “

Point 6: We have addressed reviewer 1’s comments in the discussion.

Reviewer’s comments

Changes (in red) in the manuscript revision

Otherwise, our measurement tool was already developed and examined on small population of pregnant woomen in Taiwan [17-18]. A factor analysis approach for a large sample to construct their robust validity future.

The cited references are not appropriate as the first sentence above is referring to previous research using the assessment tool. Please provide the appropriate reference(s).

Women (woomen) is misspelled

Finally, it is unclear what the authors are trying to communicate with the second sentence. Please clarify

Line: 378-382

We have re-written our text.

“Although we recommend the convenient measurement tool of fair psychometrics in this study, developed and examined among pregnant women in Taiwan [17-18], a further robust examination is required such as factor analyses for constructing the validity of pregnant women’s symptom distress in a larger, cross-cultural sample.”

Response to the Reviewer’s Comments

Thank you for your valuable time. On behalf of the authors, I have provided point-by-point explanations below to respond to your comments. 

Point 1: In general, the reviewer gave positive feedbacks, “Overall, the authors did an excellent job in address my concerns.”

Point 2: The reviewer kindly provided his/her opinion of specifying the number of obstetricians and participating hospitals in methods section.

Reviewer’s comments

Changes (in red) in the manuscript revision

1. Hospitals

In their response to this issue the authors indicate that the sample was drawn from 2 teaching hospitals. This information should be clearly stated in the Methods. Moreover, in their response they indicate that the sample was drawn from more than 20 obstetricians from these hospitals. It would be best if the specific number of obstetricians is clearly stated.

We have re-written relative texts in the abstract and method section, as fellow:

1)    Line 20:

“…, from the prenatal wards of two teaching hospitals in northern Taiwan.”

2)    Line 82-83:

” …,from the prenatal outpatient departments with a total of 28 obstetricians of two teaching hospitals in Taipei, Taiwan.”

Point 3: we have addressed the reviewer’s comments, mainly in the comparison to the general Taiwanese population

Reviewer’s comments

Changes (in red) in the manuscript revision

2. A brief comparison to the general Taiwanese population (particular against childbearing women) is necessary

I think there might have been some confusion about what I was asking. My question was whether the women who participated in this study differed from the general population in any significant manner. For example, were the women in this sample better educated, older/younger, had previous pregnancies, ... than the general population. Knowing whether the sample is representative of the general population is vital before general statements about the women's physical and psychological experiences can be drawn.

This is particularly important since the sample was obtained from two teaching hospital, which I am assuming are located in urban and not rural areas. Please describe how this study sample is similar to and different from the general childbearing population of Taiwan.

Line 365-368:

In order to answer the question regarding the clarify the reviewer’s queries, we added a brief comparison our sample and general Taiwanese population, as follow:

“Since our sample were mostly employed, they were relatively older, higher educated with better family income than the general population, compared to the 18,312 Taiwanese women included in the recent Taiwan Birth Cohort Study (TBCS) data bank (Chen, 2020).”

Reference:

Chen, C.-N.; Yu, H.-C.; Chou, A.-K. Association between Maternal Pre-pregnancy Body Mass Index and Breastfeeding Duration in Taiwan: A Population-Based Cohort Study. Nutrients 2020, 12, 2361. https://doi.org/10.3390/nu12082361

Point 4: We have addressed the reviewer’s comments in the result section.

Reviewer’s comments

Changes (in red) in the manuscript revision

From the most noticeable changes pattern of “Decrease then increase “ and the turn point of increasing of total, physical and psychological aspects 201 was at 22.02, 21.01and 22.34 week, respectively (Table 2).

This sentence is difficult to interpret. And it only deals with one pattern of change. Should the sentence read:

For the Decrease then increase trajectory, while the overall pattern changed at 22.02 weeks, physical distress levels increased earlier (21.01 weeks) relative to psychological distress levels (22.34 weeks).

We have re-written our text, according reviewer’s comments.

LINE:201-203

“For the Decrease then increase trajectory, while the overall pattern changed at 22.02 weeks, physical distress levels increased earlier (21.01 weeks) relative to psychological distress levels (22.34 weeks).” 

Point 5: We have addressed the reviewer’s comments in the discussion section.

Reviewer’s comments

Changes (in red) in the manuscript revision

Please provide a better explanation for what the authors think this means. Do physical factors drive the psychological distress level changes. If yes, why should we be concerned about this pattern of change.

Is the same pattern (physical before psychological) seen in the Increase then Decrease group? If yes, what does this mean. If no, why does it mean?

In order to answer the question, we added a brief description in the discussion section as follow:

1). Line 265-271:

“We also found that physical symptom distress changed prior to the psychological distress. This pattern has signified the importance that health care providers may prevent both types of symptom distress by first relieving the physical symptoms. In particular, education on self-care could be provided emphasized on alleviating physical symptoms in the first trimester. Early reduction of physical symptom distress would be key to help with increased levels of perceived psychological and overall symptom distress”.

2). Line 239-242:

“The key cause of increased physical, and later psychological, symptom distress may be originated from maternal physiological changes of fetal growth and increasing uterine size at 21-24 gestational weeks [24,25]. ……We believe that the anatomical and physical changes have resulted in increased distress levels along with gestation weeks. Such changes not only affect physical but psychological and overall symptom distress. “

Point 6: We have addressed reviewer 1’s comments in the discussion.

Reviewer’s comments

Changes (in red) in the manuscript revision

Otherwise, our measurement tool was already developed and examined on small population of pregnant woomen in Taiwan [17-18]. A factor analysis approach for a large sample to construct their robust validity future.

The cited references are not appropriate as the first sentence above is referring to previous research using the assessment tool. Please provide the appropriate reference(s).

Women (woomen) is misspelled

Finally, it is unclear what the authors are trying to communicate with the second sentence. Please clarify

Line: 378-382

We have re-written our text.

“Although we recommend the convenient measurement tool of fair psychometrics in this study, developed and examined among pregnant women in Taiwan [17-18], a further robust examination is required such as factor analyses for constructing the validity of pregnant women’s symptom distress in a larger, cross-cultural sample.”

Response to the Reviewer’s Comments

Thank you for your valuable time. On behalf of the authors, I have provided point-by-point explanations below to respond to your comments. 

Point 1: In general, the reviewer gave positive feedbacks, “Overall, the authors did an excellent job in address my concerns.”

Point 2: The reviewer kindly provided his/her opinion of specifying the number of obstetricians and participating hospitals in methods section.

Reviewer’s comments

Changes (in red) in the manuscript revision

1. Hospitals

In their response to this issue the authors indicate that the sample was drawn from 2 teaching hospitals. This information should be clearly stated in the Methods. Moreover, in their response they indicate that the sample was drawn from more than 20 obstetricians from these hospitals. It would be best if the specific number of obstetricians is clearly stated.

We have re-written relative texts in the abstract and method section, as fellow:

1)    Line 20:

“…, from the prenatal wards of two teaching hospitals in northern Taiwan.”

2)    Line 82-83:

” …,from the prenatal outpatient departments with a total of 28 obstetricians of two teaching hospitals in Taipei, Taiwan.”

Point 3: we have addressed the reviewer’s comments, mainly in the comparison to the general Taiwanese population

Reviewer’s comments

Changes (in red) in the manuscript revision

2. A brief comparison to the general Taiwanese population (particular against childbearing women) is necessary

I think there might have been some confusion about what I was asking. My question was whether the women who participated in this study differed from the general population in any significant manner. For example, were the women in this sample better educated, older/younger, had previous pregnancies, ... than the general population. Knowing whether the sample is representative of the general population is vital before general statements about the women's physical and psychological experiences can be drawn.

This is particularly important since the sample was obtained from two teaching hospital, which I am assuming are located in urban and not rural areas. Please describe how this study sample is similar to and different from the general childbearing population of Taiwan.

Line 365-368:

In order to answer the question regarding the clarify the reviewer’s queries, we added a brief comparison our sample and general Taiwanese population, as follow:

“Since our sample were mostly employed, they were relatively older, higher educated with better family income than the general population, compared to the 18,312 Taiwanese women included in the recent Taiwan Birth Cohort Study (TBCS) data bank (Chen, 2020).”

Reference:

Chen, C.-N.; Yu, H.-C.; Chou, A.-K. Association between Maternal Pre-pregnancy Body Mass Index and Breastfeeding Duration in Taiwan: A Population-Based Cohort Study. Nutrients 2020, 12, 2361. https://doi.org/10.3390/nu12082361

Point 4: We have addressed the reviewer’s comments in the result section.

Reviewer’s comments

Changes (in red) in the manuscript revision

From the most noticeable changes pattern of “Decrease then increase “ and the turn point of increasing of total, physical and psychological aspects 201 was at 22.02, 21.01and 22.34 week, respectively (Table 2).

This sentence is difficult to interpret. And it only deals with one pattern of change. Should the sentence read:

For the Decrease then increase trajectory, while the overall pattern changed at 22.02 weeks, physical distress levels increased earlier (21.01 weeks) relative to psychological distress levels (22.34 weeks).

We have re-written our text, according reviewer’s comments.

LINE:201-203

“For the Decrease then increase trajectory, while the overall pattern changed at 22.02 weeks, physical distress levels increased earlier (21.01 weeks) relative to psychological distress levels (22.34 weeks).” 

Point 5: We have addressed the reviewer’s comments in the discussion section.

Reviewer’s comments

Changes (in red) in the manuscript revision

Please provide a better explanation for what the authors think this means. Do physical factors drive the psychological distress level changes. If yes, why should we be concerned about this pattern of change.

Is the same pattern (physical before psychological) seen in the Increase then Decrease group? If yes, what does this mean. If no, why does it mean?

In order to answer the question, we added a brief description in the discussion section as follow:

1). Line 265-271:

“We also found that physical symptom distress changed prior to the psychological distress. This pattern has signified the importance that health care providers may prevent both types of symptom distress by first relieving the physical symptoms. In particular, education on self-care could be provided emphasized on alleviating physical symptoms in the first trimester. Early reduction of physical symptom distress would be key to help with increased levels of perceived psychological and overall symptom distress”.

2). Line 239-242:

“The key cause of increased physical, and later psychological, symptom distress may be originated from maternal physiological changes of fetal growth and increasing uterine size at 21-24 gestational weeks [24,25]. ……We believe that the anatomical and physical changes have resulted in increased distress levels along with gestation weeks. Such changes not only affect physical but psychological and overall symptom distress. “

Point 6: We have addressed reviewer 1’s comments in the discussion.

Reviewer’s comments

Changes (in red) in the manuscript revision

Otherwise, our measurement tool was already developed and examined on small population of pregnant woomen in Taiwan [17-18]. A factor analysis approach for a large sample to construct their robust validity future.

The cited references are not appropriate as the first sentence above is referring to previous research using the assessment tool. Please provide the appropriate reference(s).

Women (woomen) is misspelled

Finally, it is unclear what the authors are trying to communicate with the second sentence. Please clarify

Line: 378-382

We have re-written our text.

“Although we recommend the convenient measurement tool of fair psychometrics in this study, developed and examined among pregnant women in Taiwan [17-18], a further robust examination is required such as factor analyses for constructing the validity of pregnant women’s symptom distress in a larger, cross-cultural sample.”

Reviewer 2 Report

Dear authors,

Thank you for addressing my comments.

Kind regards

Author Response

Thank you.